# Felodipine Determination by a CdTe Quantum Dot-Based Fluorescent Probe

**DOI:** 10.3390/mi13050788

**Published:** 2022-05-18

**Authors:** Yuguang Lv, Yuqing Cheng, Kuilin Lv, Guoliang Zhang, Jiang Wu

**Affiliations:** 1School of Stomatology, Jiamusi University, Jiamusi 154002, China; zglhsy@163.com; 2College of Pharmacy, Jiamusi University, Jiamusi 154002, China; zrx080802@163.com; 3State Key Laboratory of Green Building Materials, China State Building Materials Research Institute Co., Ltd., Beijing 100000, China; lvkuilin@ctc.ac.cn

**Keywords:** CdTe quantum dots, felodipine, fluorescence quenching, drug testing

## Abstract

In this work, a CdTe quantum dot-based fluorescent probe was synthesized to determine felodipine (FEL). The synthesis conditions, structure, and interaction conditions with FEL of CdTe quantum dots were analysed by fluorescence spectrophotometry, Fourier transform infrared spectroscopy (FT-IR), X-ray diffraction (XRD), UV–visible spectroscopy, and TEM. The CdTe QD concentration was 2.0 × 10^−4^ mol/L. The amount of quantum dots controlled in the experiment was 0.8 mL. The controlled feeding ratio of N (Cd^2+^):N (Te^2−^):N (TGA) was 2:1:4, the heating temperature was 140 °C, the heating time was 60 min, and the pH of the QD precursor was adjusted to 11 for subsequent experiments. The UV–visible spectrum showed that the emission wavelength of CdTe quantum dots at 545 nm was the strongest and symmetric. The particle size of the synthesized quantum dots was approximately 5 nm. In the interaction of CdTe quantum dots with FEL, the FEL dosage was 1.0 mL, the optimal pH value of Tris-HCl buffer was 8.2, the amount of buffer was 1.5 mL, and the reaction time was 20 min. The standard curve of FEL was determined under the optimal synthesis conditions of CdTe quantum dots and reaction of CdTe quantum dots with FEL. The linear equation was Y = 3.9448x + 50.068, the correlation coefficient R2 was 0.9986, and the linear range was 5 × 10^−6^–1.1 × 10^−4^ mol/L. A CdTe quantum dot-based fluorescent probe was successfully constructed and could be used to determine the FEL tablet content.

## 1. Introduction

Hypertension is universally recognized as one of the significant reasons for cardiovascular illnesses, and may prompt unpredictable changes in fat and sugar digestion, actuate the distress of significant objective organs and, in any event, jeopardize the survival of patients [1,2]. There are many types of commonly used drugs for hypertension. Among the various drugs, calcium channel antagonists mainly fall into three categories [3], among which dihydropyridine drugs with the strongest cardiovascular selectivity are used the most [4]. Compared with other types of blood pressure-lowering drugs, calcium channel antagonists do not impact the regular blood glucose and lipid metabolism of patients and do not have particularly visible side effects [5]. In conclusion, calcium channel antagonists are very suitable for the treatment of hypertension with metabolic syndrome, especially for patients with ACEI intolerance. These antagonists are also suitable for the treatment of diabetes, angina pectoris and other diseases [6].

FEL tablets are highly active and can selectively stretch and dilate peripheral arterioles, which can significantly reduce peripheral vascular resistance and blood pressure of the entire body [7]. The selectivity of drugs for blood vessels is very high. In general, normal doses of the drug lower blood pressure well and have no significant effect on all aspects of cardiac function [8]. FEL can simultaneously reduce the blood pressure of patients and significantly increase the plasma auxin concentrations [9]. The elimination effect of FEL on the liver head is very obvious. To control the blood pressure of patients within the range of normal human blood pressure within 24 h after medication to better and more safely defend target organs and avoid sudden death due to a rapid increase in blood pressure [10,11], it is best to take medication only once a day.

Quantum dots (QDs) are semiconductor nanocrystals. To obtain QDs of different sizes, only the synthesis conditions need to be changed. If the particle size changes, the photoelectric performance of QDs will also change [12,13]. In recent years, due to the small size, high fluorescence intensity, good light stability and other characteristics of QDs, research on the combination of quantum dots with biometric molecules has become mainstream. However, quantum dots also have strong biological toxicity, which greatly affects their application [14]. CdTe QDs have excellent fluorescence performance, and they have a wide and continuous excitation spectrum, a narrow and symmetrical emission spectrum, high quantum yield and strong fluorescence performance, which makes them suitable for fluorescence sensors and biological imaging [15,16]. Various studies have shown that QDs with functional modification can be used for immunofluorescence detection and labelling of living samples, especially in the field of drug analysis [17]. In this paper, QDs were used as a fluorescence probe to determine the content of FEL tablets in vitro.

At present, few studies have been reported on the use of RP-HPLC [18], UV–VIS [19], and HPTLC [20] for the determination of FEL in bulk drugs or pharmaceutical preparations. Although UV–visible spectrophotometry has the advantages of simple operation, the accuracy cannot satisfy the requirements of experiments, and the sample concentration has high requirements, which are relatively difficult to attain, and the volumetric analysis is poor in sensitivity. HPLC has high stability and specificity, low detection and quantification limits, and is suitable for the quantitative determination of trace lodipine and its related substances [21,22]. Emam Aml A et al. [23] used HPLC to determine the content of FEL in human plasma. HPLC has the advantages of high detection efficiency, but it cannot detect insoluble samples, and the cost of sample analysis is relatively high. Currently, the fluorescence probe method has attracted the attention of researchers with a wide range of normally involved techniques to examine the effectiveness, responsiveness, and repeatability, but there are few reports on the determination of FEL tablets by the fluorescence probe method. The detection method needs to be improved from the aspects of analysis efficiency, sensitivity and repeatability to expand the analysis range, improve the analysis efficiency and reduce sample consumption [24].

The main objective of this experiment is to study FEL by CdTe quantum spot fluorescence spectrometry and to provide a new idea and method for the simple and accurate determination of FEL in tablets.

## 2. Materials and Methods

### 2.1. Synthesis of CdTe Quantum Dots

First, 0.120 g of CdCl_2_·2.5 H_2_O and 72.7 μL of thioglycolic acid (TGA) (in 200 mL of high-purity water) were added to a round-bottom flask, the pH value was adjusted to approximately 11.0 with 1.00 mol/L NaOH solution, and high-purity nitrogen was passed through the solution for 30 min. Then, 210 μL of a newly prepared sodium telluride solution was added under nitrogen protection. The colour of the solution immediately turned yellow, and the CdTe quantum dot precursor was obtained. The original solution was transferred to a hydrothermal synthesis reactor lined with Teflon and heated in an oven at 140 °C for different times to obtain transparent CdTe quantum dot solutions of different colours [25,26].

### 2.2. Influence of Synthesis Conditions on CdTe Quantum Dots

Different concentrations of CdTe QD solutions (5 × 10^−5^ mol/L, 1.0 × 10^−4^ mol/L, 1.5 × 10^−4^ mol/L, 2.0 × 10^−4^ mol/L, 2.5 × 10^−4^ mol/L, 3.0 × 10^−4^ mol/L, and 3.5 × 10^−4^ mol/L) were screened to explore the fluorescence intensity of the system (F2500, Hitachi Manufacturing Co., Ltd.,Schaumburg, IL, USA).

We selected dosages of 0.2 mL, 0.4 mL, 0.6 mL, 0.8 mL, 1.0 mL, 1.2 mL, and 1.4 mL of CdTe QDs at the same concentration, which influenced the fluorescence intensity of the system, and the best dosage of CdTe QDs was determined. When heated at 100 °C, 120 °C, 140 °C, and 160 °C for the same duration, the fluorescence intensity of CdTe QDs was determined from the fluorescence emission spectra. The best heating time was determined from the fluorescence emission spectra of CdTe quantum dots at different reaction times (30 min, 60 min, 90 min, and 120 min). The controlled feeding ratio of N (Cd^2+^):N (Te^2−^):N (TGA) was 2:1:4, the heating temperature was 140 °C, the heating time was 60 min, and the effect of pH values (8, 9, 10, 11, and 12) of the reaction precursor solution on the fluorescence properties of CdTe quantum dots was analysed.

### 2.3. Structural Characterization of CdTe Quantum Dots

The UV–visible spectrum (UV/Vis-265) was obtained for CdTe QDs by regulating the concentration of quantum dots at 1.0 × 10^−4^ mol/L, in the wavelength range of 800–200 nm, and the absorption spectrum was determined by quartz cuvette absorbance.

The concentration of the quantum dot solution was adjusted to 1.0 × 10^−4^ mol/L, and the fluorescence spectrum was detected by a fluorescence spectrophotometer (970 CRT).

The appropriate amount of methanol was added to the CdTe solution, and fully mixed by ultrasound. The solution was centrifuged with a high-speed centrifuge and the bottom precipitate was taken, washed with methanol and centrifuged three times. The precipitate was dried in a vacuum drying oven. The prepared quantum dot solution was evaporated to 1/5 of its original volume by a rotary evaporator. After centrifugation and washing, the solution was vacuum dried at 30 °C for 6 h. The CdTe QDs were characterized by Fourier transform infrared spectroscopy (Nexus-470), and the resolution was better than 0.4 cm^−1^. The CdTe QD powder obtained after centrifugal drying was characterized by X-ray diffraction (BRUKER D8 ADVANCE) with a Cu target Kα-ray tube (λ = 0.1541 nm) at 40 kV and a scanning range of 10–80°. The QDs obtained after centrifugal drying were dispersed into anhydrous ethanol solution (0.001–0.01 mg/mL) by ultrasound or a cell crusher until the particles were invisible to the naked eye. A pipette gun was used to drop 40 µL of the dispersion onto the front of a TEM copper mesh in two batches, which was dried at room temperature for TEM observation (HITACHI-800).

### 2.4. Interaction of CdTe Quantum Dots with FEL

The fluorescence spectrum of the system with the addition of FEL was analysed by a fluorescence spectrophotometer. The effect of the FEL dosage on the fluorescence quenching degree was investigated. FEL cannot stably exist in strong acid and alkali environments, and it is easier to decompose in strong acid environments. Therefore, this experiment used the weak alkaline buffer Tris-HCl to investigate the quenching degree of the fluorescence intensity of the system when the pH value of the buffer was 7.3, 7.6, 7.9, 8.2, 8.5, 8.8 and 9.1. The effect of the amount of buffer solution on the fluorescence quenching degree of the system was investigated. The quenching of the fluorescence intensity of the system for different reaction times was detected.

## 3. Results and Discussion

### 3.1. Influence of Synthesis Conditions on CdTe Quantum Dots

For the molar ratio of Cd^2+^ to Te^2−^, Te^2−^ is highly sensitive to O_2_, and Te^2−^ is easily oxidized, which will decrease the fluorescence intensity of CdTe quantum dots. Excessive Cd^2+^ can inhibit the oxidation of Te^2−^, and n(Cd^2+^):n(Te^2−^) is controlled at 2:1. The molar ratio of NaBH_4_ to Te^2−^ is important since NaBH_4_ is active and can react with part of the deionized water, which implies that the amount of reaction with Te will be reduced. To improve the conversion rate of Te, the amount of NaBH_4_ can be doubled, which requires n(NaBH_4_):N (Te^2−^) = 2:1. The molar ratio of Cd^2+^ to TGA and the ratio of Cd^2+^ to TGA greatly affect the activity of quantum dots. When the amount of TGA is low, CdTe quantum dots easily precipitate, slowly grow and have weak antioxidant ability. In this experiment, n(Cd^2+^):N (TGA) was controlled at 1:2.

The concentration of quantum dots greatly affects the sensitivity and linear range of the method. The fluorescence differences of quantum dot solutions with different concentrations were investigated. As shown in Figure 1A, the fluorescence quenching degree of the system increased with increasing CdTe QD concentration, and the fluorescence quenching degree (225.338 ± 0.15326, *n* = 3) was the most obvious at 2.0 × 10^−4^ mol/L. When the concentration continued to increase, a self-bursting phenomenon occurred, which weakened the fluorescence burst intensity. Therefore, 2.0 × 10^−4^ mol/L CdTe QDs were used for the experiments (the concentration of quantum dots was based on Cd^2+^).

A single variable was controlled to explore the effect of CdTe QD dosage on the fluorescence quenching degree of the system. As shown in Figure 1B, the fluorescence quenching degree of the system increased with increasing dosage of CdTe QDs. When the dosage was 0.8–1 mL, the fluorescence quenching degree (241.406 ± 0.2267, *n* = 3) was strong and stable; when the dosage was more than 1.0 mL, the quenching degree was weakened. Therefore, the amount of quantum dots controlled in the experiment was 0.8 mL.

When heated at 140 °C for the same duration, the fluorescence intensity of CdTe QDs was the strongest, and the emission peak was wide and symmetrical (Figure 1C).

When heated at 140 °C for 60 min, the fluorescence intensity of CdTe QDs was the strongest, and the emission peak was wide and symmetrical. Therefore, quantum dots heated for 60 min were selected for subsequent experiments (Figure 1D).

When the controlled feeding ratio of N (Cd^2+^):N (Te^2−^):N (TGA) was 2:1:4, the heating temperature was 140 °C, the heating time was 60 min, the effect of pH values (8, 9, 10, 11, 12) of the reaction precursor solution on the fluorescence properties of CdTe quantum dots was investigate. With increasing pH, the fluorescence intensity gradually increased, and the fluorescence intensity was the strongest when pH = 11 and subsequently exhibited a decreasing trend. Therefore, the pH of the QD precursor was adjusted to 11 for subsequent experiments.

### 3.2. Structural Characterization of CdTe Quantum Dots

Figure 2A,B show that the UV–visible spectrum and the fluorescence spectrum obtained for CdTe quantum dots heated at 140 °C for 60 min, and the emission wavelength of CdTe quantum dots at 545 nm was the strongest and symmetric.

From the transmission electron microscopy of CdTe quantum dots, the particle size of the synthesized quantum dots was approximately 5 nm, and the size was uniform (Figure 2C).

Figure 2D shows that 3989 cm^−1^ was the absorption peak of -OH, which might be caused by incomplete drying of methanol; 683 cm^−1^ was the stretching vibration of C-S; 1542 cm^−1^, 1403 cm^−1^ and 769 cm^−1^ correspond to the anti-symmetric stretching vibration, symmetric stretching vibration and deformation vibration of COO^−^, respectively. The absence of an S-H stretching vibration peak at 2558 cm^−1^ indicates that TGA was modified by S atoms via covalent bonds on the surface of CdTe quantum dots (Figure 2E).

Figure 2F shows that three peaks were located at 24.96°, 41.98° and 48.15°, which correspond to the (111), (220) and (311) crystal planes, respectively, of the cubic crystal system. The broadening of the peaks indicates that the size of the synthesized quantum dots was very small.

### 3.3. Interaction of CdTe Quantum Dots with FEL

Figure 3 shows the fluorescence spectrum of the system with the addition of FEL. The quantum dots have strong fluorescence intensity, and the fluorescence intensity decreased after FEL was added. The fluorescence intensity of the system decreased with increasing FEL concentration, which indicates that FEL produced fluorescence quenching for the CdTe quantum dots, but the position of the emission peak did not change. In addition, there was a linear relationship between the decrease in fluorescence intensity and the FEL concentration. Therefore, this experimental method can be used as the basis to determine the FEL content via fluorescence methods.

The effect of the FEL dosage on the fluorescence quenching degree was investigated [27,28]. As shown in Figure 4A, the fluorescence quenching degree increased with increasing FEL dosage. The fluorescence quenching degree of the system reached the maximum (245.164 ± 0.0375, *n* = 3) when the dosage was 1.0 mL and tended to weaken when the dosage exceeded 1.0 mL. Therefore, the FEL dosage was controlled to be 1.0 mL in the experiment.

FEL cannot stably exist in strong acid and alkali environments, and it is easier to decompose in strong acid environments. Therefore, this experiment used the weakly alkaline buffer Tris-HCl to investigate the quenching degree of the fluorescence intensity of the system when the pH value of the buffer was 7.3, 7.6, 7.9, 8.2, 8.5, 8.8 and 9.1. As shown in Figure 4B, the fluorescence quenching degree (262.335 ± 0.0375, *n* = 3) is the strongest when the pH is 8.2, so the optimal pH value of Tris-HCl buffer is 8.2. The carboxyl oxygen atoms and Cd atoms in GA on the surface of quantum dots can cooperate twice to form a ring substance, which enhances the fluorescence performance of the system (Figure 5). In addition, pH can affect the structure of the substance.

The effect of the amount of buffer solution on the fluorescence quenching degree of the system was investigated. As shown in Figure 4C, the fluorescence quenching degree increased with the increase in the dosage of buffer solution, and the fluorescence quenching degree was strong and stable with a dosage of 1.2–2.0 mL. When 1.5 mL of buffer solution was used in the experiment, the fluorescence intensity was 258.708 ± 0.02003 (*n* = 3). Therefore, when the dosage continued to increase, the fluorescence quenching degree weakened.

The quenching of the fluorescence intensity of the system under different reaction times was detected. As shown in Figure 4D, the fluorescence quenching degree (247.768 ± 0.01682, *n* = 3) was the strongest at approximately 20 min and tended to be stable within 30 min. Therefore, the experiment started after 20 min of reaction. 

The standard curve of FEL was determined under the optimal synthesis conditions of CdTe quantum dots and reaction of CdTe quantum dots with FEL. The linear equation is Y = 3.9448x + 50.068, the correlation coefficient R2 is 0.9986, and the linear range is 5 × 10^−6^–1.1 × 10^−4^ mol/L (Figure 6). In the literature, the micellar liquid chromatography-fluorescence detection method for the determination of FEL in tablets exhibited a linear range of 0.05–15 ug/mL, with a lower LOD of 0.011 ug/mL and a lower LOQ of 0.032 ug/mL [29]. FEL can interact with quantum dots, which can quench the fluorescence intensity of the system. The results show a good linear relationship between the decrease in fluorescence intensity and the concentration of FEL in a specific range, which formed the basis of fluorimetry to determine the FEL content.

### 3.4. Methodological Investigation

#### 3.4.1. Precision

Table 1 shows the precision results of FEL. The RSD value of felodipine was 1.36%, which indicates that the precision performance of this method was good.

#### 3.4.2. Accuracy

The recovery results are shown in Table 2. The RSD values of three different concentrations of the FEL reserve solution were 1.9%, 1.6% and 1.4%, all of which were less than 2.00%. The accuracy of this method was good.

#### 3.4.3. Influence of Coexisting Substances

According to the actual situation of possible coexistence, the influence of various interfering substances (common ions and other common antihypertensive drugs) was investigated on the determination of 2.0 × 10^−5^ mol/L felodipine. Na^+^, Cu^2+^, Ca^2+^, K^+^, propranolol, hydrochlorothiazide, metoprolol and valsartan were selected. As shown in Table 3, 50 times of Na^+^, Cu^2+^, Ca^2+^ and K^+^ at 50 times the FEL concentration had almost no effect on the determination results. Propranolol and valsartan at 10 times the FEL concentration had almost no effect on the results. However, hydrochlorothiazide and metoprolol at 10 times the FEL concentration had some effect on the results.

#### 3.4.4. Sample Content Determination

There was no significant difference between the results of the detection method in this study and the labelled amount of FEL tablets. All relative standard deviations of the samples were within the allowable range, which indicates that the method to construct the FEL content in FEL tablets in this study was accurate and reliable. The content determination data of the samples were recorded as follows (in Table 4).

#### 3.4.5. Preliminary Study on the Quenching Mechanism

Fluorescence quenching occurred when the fluorescence intensity was reduced by physical and chemical interactions between fluorescent substances and quenching agents. At the binding site, hydrogen bonding, electrostatic forces and van der Waals forces impacted the fluorescence properties of QDs.

The reason for this phenomenon is the large number of carboxyl functional groups on the surface of CdTe QDs, and the strong electronegativity of O atoms can combine with H^+^ generated by –COOH ionization to form hydrogen bonds and complexes with phlodidine. These substances are wrapped on the surface of CdTe QDs for electron transfer, which results in fluorescence quenching. Figure 7 shows a simulation diagram of the interaction between FEL and CdTe quantum dots.

## 4. Conclusions

In this study, a CdTe quantum dot-based fluorescent probe was successfully constructed, which can be used to detect the FEL content. There was no substantial difference between the results of the determination method and the labelled content of felodipine tablets. Therefore, this method can be used to determine the content of FEL in drugs and different substances. It provides an accurate and reliable method to detect felodipine. This method has the characteristics of simple reaction conditions, easy control, simple and quick operation, and high sensitivity.

## Figures and Tables

**Figure 1 micromachines-13-00788-f001:**
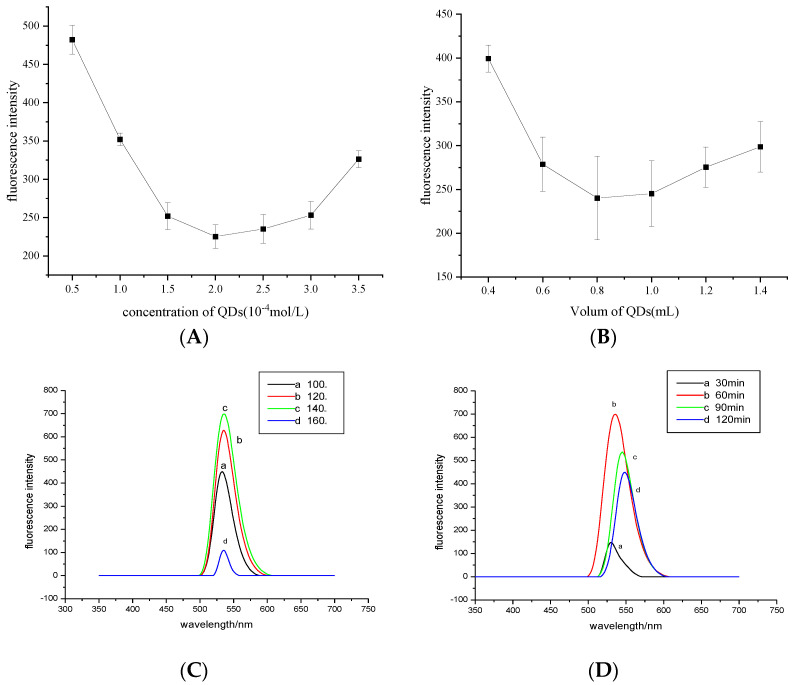
Influence of (**A**) the quantum dot concentration; (**B**) the amount of CdTe QDs; (**C**) different heating temperatures; and (**D**) different reaction times.

**Figure 2 micromachines-13-00788-f002:**
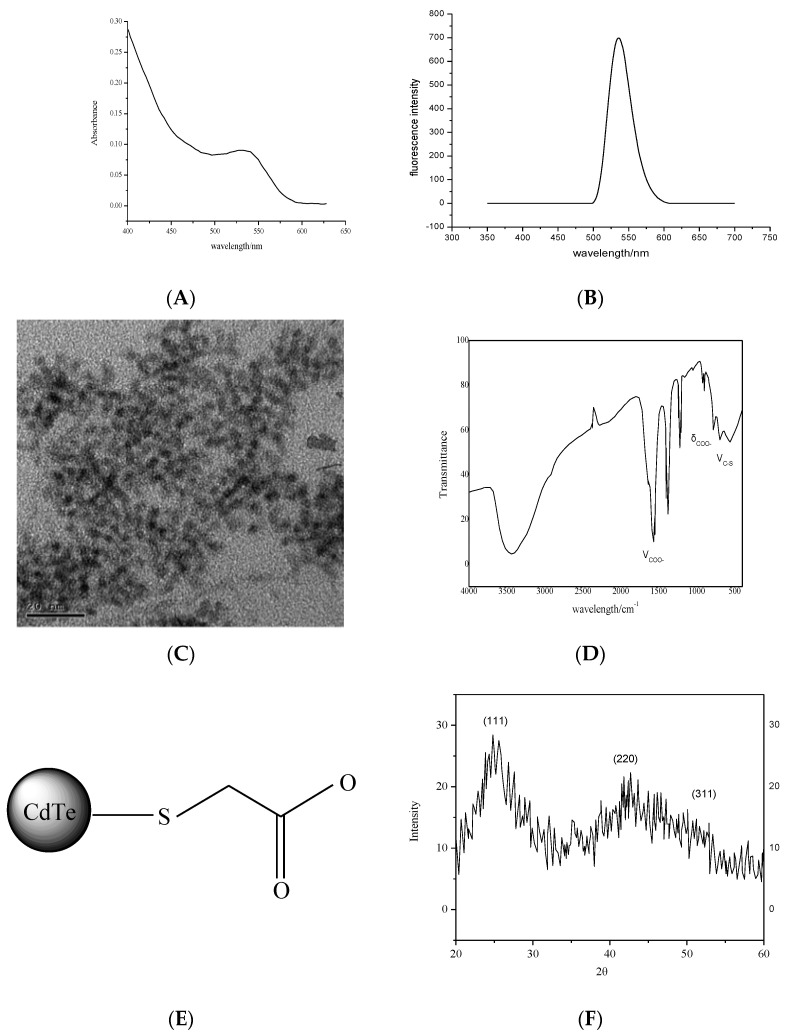
Structural characterization of CdTe quantum dots. (**A**) Ultraviolet absorption spectra of CdTe quantum dots. (**B**) Fluorescence spectra of CdTe quantum dots. (**C**) Transmission electron microscopy of CdTe quantum dots. (**D**) Infrared spectra of CdTe quantum dots. (**E**) Schematic diagram of surface structure of quantum dots modified by thioglycolic acid. (**F**) XRD pattern of quantum dots.

**Figure 3 micromachines-13-00788-f003:**
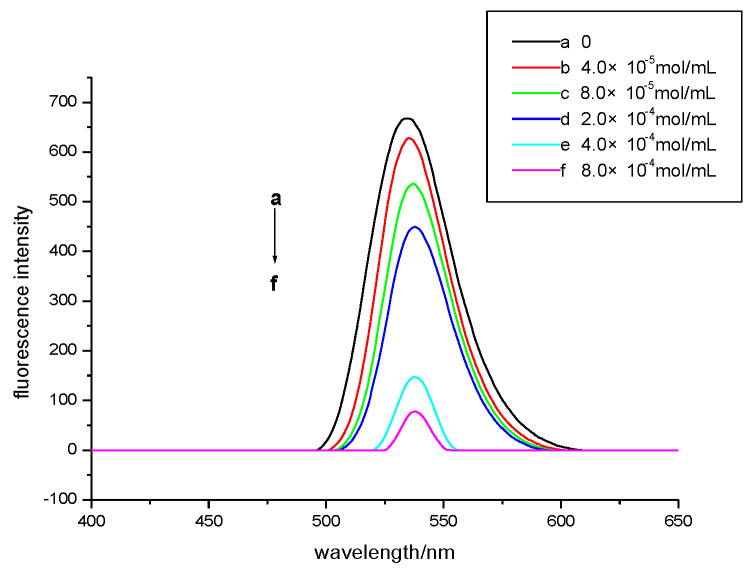
Fluorescence spectra of different concentrations of FEL interacting with CdTe QDs.

**Figure 4 micromachines-13-00788-f004:**
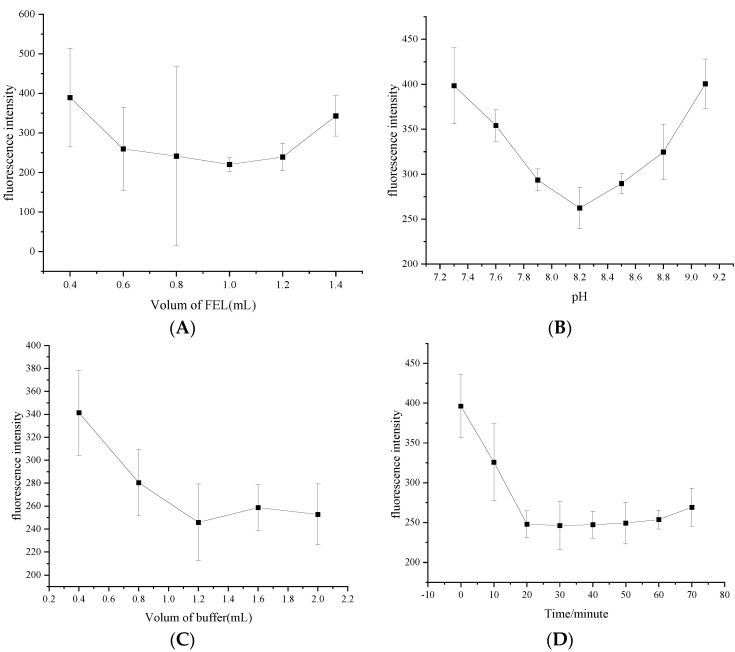
Effects of (**A**) FEL dosage; (**B**) pH of the buffer solution; (**C**) amount of buffer solution; and (**D**) reaction time.

**Figure 5 micromachines-13-00788-f005:**
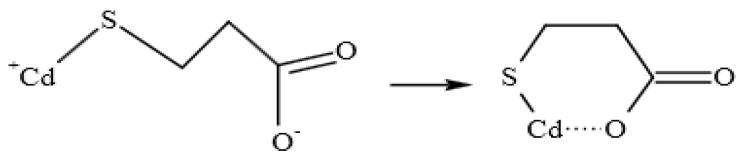
The structure of the complex formed by thioglycolic acid and Cd^2+^.

**Figure 6 micromachines-13-00788-f006:**
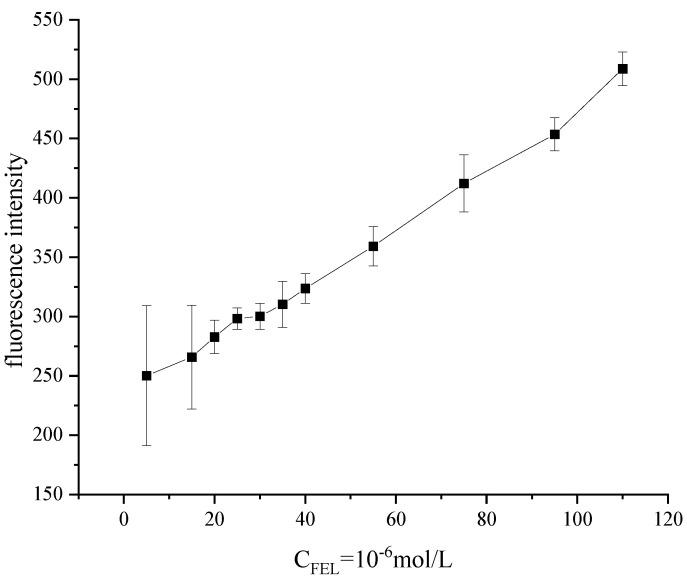
Standard working curve.

**Figure 7 micromachines-13-00788-f007:**
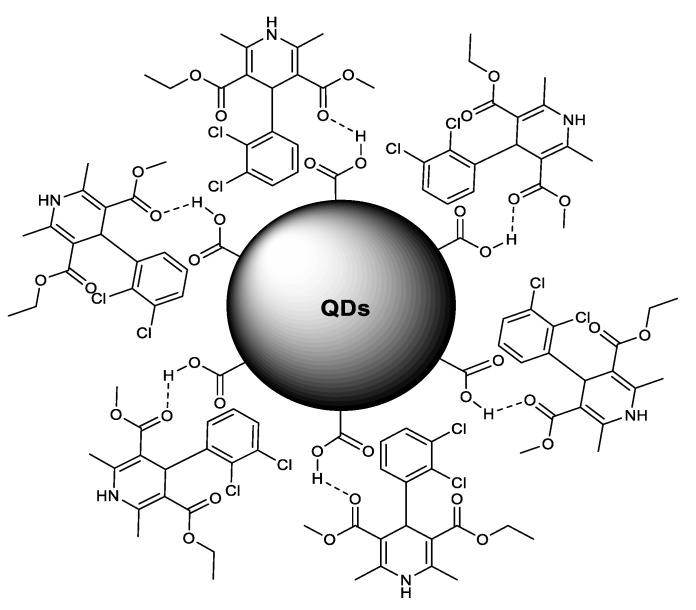
Simulation diagram of the interaction between FEL and CdTe QDs.

**Table 1 micromachines-13-00788-t001:** Precision test results of FEL detection.

Number of Measurements	ΔF (F − F_0_)	ΔF Average Value	RSD (%)
1	475.28		
2	483.65		
3	466.32		
4	470.15	472.88	1.36
5	474.59		
6	467.28		

**Table 2 micromachines-13-00788-t002:** Test results of the FEL detection accuracy.

Concentration (10^−6^ mol/L)	Recovery Mean ± SD/%	RSD/%
0.5	99.7 ± 1.92	1.9
3.0	99.5 ± 1.63	1.6
6.0	101.1 ± 1.45	1.4

**Table 3 micromachines-13-00788-t003:** Influence of coexisting substances.

Interfering Substance	Interfering Substance (mol/L)	RSD (%)	Interfering Substance	Interfering Substance (mol/L)	RSD (%)
Na^+^	100	0.8	propranolol	10	0.8
Cu^2+^	100	0.7	hydrochlorothiazide	10	2.3
Ca^2+^	100	0.9	metoprolol	10	5.4
K^+^	100	0.7	valsartan	10	0.9

**Table 4 micromachines-13-00788-t004:** Determination of samples and comparison with the labelled amount.

Number	Sample Measurement (mg/L)	Average Measured Value Mean ± SD (mg/L)	Labelled Amount (mg/L)	RSD (%)
1	98.4, 99.1, 100.9	99.467 ± 1.290	100	1.29
2	99.2, 103.6, 97.7	100.167 ± 3.066	100	0.306

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
