# Peer review of "Felodipine Determination by a CdTe Quantum Dot-Based Fluorescent Probe"

_micromachines, 2022, doi:10.3390/mi13050788_

Round 1

Reviewer 1 Report

Review comments on micromachines-1710222: Felodipine by CdTe quantum spot fluorescence spectrometry

The manuscript described the evaluation of CdTe quantum dots on the quantitation of felodipine. The authors provided some data to highlight and interpret the interaction between CdTe quantum dots and felodipine.

In this revised version, the authors partly improved the manuscript by expanding the Introduction section, adding the Method section, and modifying the Results and discussion section. However, there are still many issues in this revised manuscript. The manuscript in the current form is still unsuitable for publication in Micromachines. The authors may consider the detailed comments below and improve the manuscript.

  1. The developed method can be applied to determine felodipine in dosage forms (e.g., tablets and capsules) or biological samples (e.g., blood and tissues). The authors should clarify the application at the beginning.
  2. From the data shown in this study, the method can only be applied to determine felodipine in dosage forms. If so, the first and second paragraphs of the Introduction section should be shortened and modified. It is better to discuss the analytical methods to determine felodipine in dosage forms in previous studies. Most of them are UV-Vis spectrophotometry and HPLC.
  3. If the authors aimed for biological samples, the LOD and LOQ of the method should be identified and compared with the conventional concentration of felodipine in human blood. The validation of the method (specificity, accuracy, precision, etc.) should be conducted using a matrix (human or animal blood/ plasma) to show its ability to apply in clinical or experimental samples. The authors should compare the LOD, LOQ, and accuracy of the developed method with those of previously reported methods. In lines 57-69, the authors mentioned that the devised method could overcome some shortcomings of previous methods. Therefore, the authors must prove that by data.
  4. The linear range is 5× 10^-6 - 1.1 ×10^-4 mol/L. The authors should compare it with previously reported methods (e.g., UV-Vis spectrophotometry, HPLC, and LC-MS).
  5. Each experiment or measurement should be repeated at least three times. Data in Figures should be means ± SDs.
  6. Section 4.1 - Structural characterization of CdTe quantum dots: the transmission electron microscopy, FTIR, TGA, and XRD data were discussed. However, these methods were not mentioned in the Method section. Notably, there were no data (figures). The authors should present the transmission electron microscopy, FTIR, TGA, and XRD data together with the discussion.
  7. Lines 57-62: the authors should include relevant references to support the statement. Current methods for felodipine determination, such as UV-Vis spectrophotometry, HPLC, and LC-MS, should be cited and discussed. Generally, LC-MS is the best option for determining felodipine in blood samples.
  8. Line 58: are UV–visible spectrophotometry and UV spectrophotometry two different methods?
  9. Lines 59-61: Please clarify which method has unsatisfied accuracy. HPLC and LC-MS are highly accurate methods.
  10. Lines 60-61: Please clarify the statement “the sample concentration has high requirements.” Does it mean that these methods have low sensitivity?
  11. The authors should state the primary goal, novelty, and contribution of this study to the field (at the end of the Introduction section).
  12. The Method section should be revised. It was not well prepared. For example, in section 2.3.1- Selection of the felodipine dosage, it was described that “The effect of the FEL dosage on the fluorescence quenching degree was investigated.” This sentence is vague and insufficient. It is required to clarify the FEL dosage and how to investigate the “fluorescence quenching degree.” Similar to other parts of the Method section.
  13. The authors should clarify the number of replications per experiment or measurement and how presented data are presented. Is there any statistical test?
  14. Sections 3 and 4 should be combined. The authors should rearrange the data and discussion.
  15. Both FEL and felodipine were used. The authors should keep using the abbreviation consistently.

Author Response

  April 28, 2022

Dear Editor,

We would like to submit the revised manuscript (micromachines-1710222) entitled “Felodipine determination by a CdTe quantum dots-based fluorescent probe” for a further review and publication in the journal Micromachines. Based on the reviewers’ comments, we have made significant revisions in a point-to-point manner (see below for details). We hope you will find that the revised manuscript now meets the standards of the journal.

Thank you very much for your consideration. We look forward to your favorable decision.

Sincerely,

Jiang Wu, Dr.

School of Stomatology, Jiamusi University, Heilongjiang 154002, China.

Email: wujiangwj0126@163.com.

Tel.: +86–454–8625581. 

Response to the comments from reviewer:

First, we would like to thank all reviewers for their critical examination of our manuscript. Their comments are very helpful for revising the manuscript. We have made significant changes to address their concerns. The issues they raised and our responses are outlined below.

Referee #1

Comment (C) 1: The developed method can be applied to determine felodipine in dosage forms (e.g., tablets and capsules) or biological samples (e.g., blood and tissues). The authors should clarify the application at the beginning.

Response (R): Thank you for your kind comments. CdTe QDs have an excellent fluorescence performance, and they have a wide and continuous excitation spectrum, a narrow and symmetrical emission spectrum, high quantum yield and strong fluorescence performance, which makes them suitable for fluorescence sensors and biological imaging [15-16]. Various studies have shown that QDs with functional modification can be used for immunofluorescence detection, labeling of living samples, and especially in the field of drug analysis [17]. In this paper, QDs were used as a fluorescence probe to determine the content of felodipine tablets in vitro. Please see line 62-69 in page 2 for details.

 “The felodipine tablets” Please see line 46 in page 1 for details.

Comment (C) 2: From the data shown in this study, the method can only be applied to determine felodipine in dosage forms. If so, the first and second paragraphs of the Introduction section should be shortened and modified. It is better to discuss the analytical methods to determine felodipine in dosage forms in previous studies. Most of them are UV-Vis spectrophotometry and HPLC.

Response (R): Thank you for your kind comments. We discuss the above problems in this paper. Please see line70-82 in page 2 for details.

At present, there are many analytical methods to determine the felodipine content, such as UV–visible spectrophotometry, volumetric analysis, liquid chromatography [18]. Although UV–visible spectrophotometry and volumetric analysis have the advantages of simple operation, the accuracy cannot satisfy the requirements of the experiment, and the sample concentration has high requirements, which are relatively difficult to attain, and the volumetric analysis is poor in sensitivity. Emam Aml A et al. [19] used HPLC to determine the content of felodipine in human plasma. HPLC and liquid chromatography-mass spectrometry have the advantages of high detection efficiency, but which cannot detect the insoluble samples, and the cost of analysis samples is relatively expensive. Currently, the fluorescence probe method has attracted the attention of researchers with a wide range of normally involved techniques to examine the effectiveness, responsiveness and repeatability, but there are not many reports on the determination of the felodipine tablets by the fluorescence probe.

Comment (C) 3: If the authors aimed for biological samples, the LOD and LOQ of the method should be identified and compared with the conventional concentration of felodipine in human blood. The validation of the method (specificity, accuracy, precision, etc.) should be conducted using a matrix (human or animal blood/ plasma) to show its ability to apply in clinical or experimental samples. The authors should compare the LOD, LOQ, and accuracy of the developed method with those of previously reported methods. In lines 57-69, the authors mentioned that the devised method could overcome some shortcomings of previous methods. Therefore, the authors must prove that by data.

Response (R): Thank you for your kind comments. We discuss the above problems in this paper. Please see line70-82 in page 2 for details.

At present, there are many analytical methods to determine the felodipine content, such as UV–visible spectrophotometry, volumetric analysis, liquid chromatography [18]. Although UV–visible spectrophotometry and volumetric analysis have the advantages of simple operation, the accuracy cannot satisfy the requirements of the experiment, and the sample concentration has high requirements, which are relatively difficult to attain, and the volumetric analysis is poor in sensitivity. Emam Aml A et al. [19] used HPLC to determine the content of felodipine in human plasma. HPLC and liquid chromatography-mass spectrometry have the advantages of high detection efficiency, but which cannot detect the insoluble samples, and the cost of analysis samples is relatively expensive. Currently, the fluorescence probe method has attracted the attention of researchers with a wide range of normally involved techniques to examine the effectiveness, responsiveness and repeatability, but there are not many reports on the determination of the felodipine tablets by the fluorescence probe.

Comment (C) 4: The linear range is 5× 10^-6 - 1.1 ×10^-4 mol/L. The authors should compare it with previously reported methods (e.g., UV-Vis spectrophotometry, HPLC, and LC-MS).

Response (R): Thank you for your kind comments. The standard curve of felodipine was determined under the optimal conditions of synthesis conditions on CdTe quantum dots and reaction of CdTe quantum dots with Felodipine.

Comment (C) 5: Each experiment or measurement should be repeated at least three times. Data in Figures should be means ± SDs.

Response (R): Thank you for your kind comments. According to the experimental data, we have modified the picture. The details are as follows.

Line 66-67 in page 4: Figure 1. Influence of (A) the quantum dot concentration; (B) the amount of CdTe QDs;

Line 236-242 in page 7: Figure 4. Effect of (A) the felodipine dosage; (B) the pH of the buffer solution; (C) the amount of buffer solution; (D) the reaction time.

Line 245-246 in page 8: Figure 6. Standard working curve

Comment (C) 6: Section 4.1 - Structural characterization of CdTe quantum dots: the transmission electron microscopy, FTIR, TGA, and XRD data were discussed. However, these methods were not mentioned in the Method section. Notably, there were no data (figures). The authors should present the transmission electron microscopy, FTIR, TGA, and XRD data together with the discussion.

Response (R): Thank you for your kind comments. This part has been adjusted in terms of materials, methods and results.

Please see line 113-118 in page 3 for details.

2.3. Structural characterization of CdTe quantum dots

The UV–visible spectrum (UV-2550) was obtained for CdTe QDs heated at 140℃ for 60 min. The particulate morphology of CdTe QDs was observed on TEM (HITACHI-800). CdTe QDs were characterized by Fourier transform infrared spectroscopy (BRUKER VERTEX70). CdTe QDs were characterized by X-ray diffraction (BRUKER D8 ADVANCE).

Please see line 168-200 in page 4-6 for details.

3.2. Structural characterization of CdTe quantum dots

Comment (C) 7: Lines 57-62: the authors should include relevant references to support the statement. Current methods for felodipine determination, such as UV-Vis spectrophotometry, HPLC, and LC-MS, should be cited and discussed. Generally, LC-MS is the best option for determining felodipine in blood samples.

Response (R): Thank you for your kind comments. We have added references. Please see line 70 in page 2 for details.

[18] Shormanov V K, Kvachakhiya L L. Felodipine distribution in organisms of warm-blooded animals. Sudebno-meditsinskaia Ekspertiza, 2020; Volume 63, pp. 47-52. doi: 10.17116/sudmed20206301147

Comment (C)8: Line 58: are UV–visible spectrophotometry and UV spectrophotometry two different methods?

Response (R): Thank you for your kind comments. This problem has been modified.

Please see line 70 in page 2 for details.

UV–visible spectrophotometry, volumetric analysis, liquid chromatography [18].

Comment (C) 9: Lines 59-61: Please clarify which method has unsatisfied accuracy. HPLC and LC-MS are highly accurate methods.

Response (R): Thank you for your kind comments. Please see line 75-77 in page 2 for details.

HPLC and liquid chromatography-mass spectrometry have the advantages of high detection efficiency, but which cannot detect the insoluble samples, and the cost of analysis samples is relatively expensive.

Comment (C) 10: Lines 60-61: Please clarify the statement “the sample concentration has high requirements.” Does it mean that these methods have low sensitivity?

Response (R): Thank you for your kind comments. Please see line 71-74 in page 2 for details.

Although UV–visible spectrophotometry and volumetric analysis have the advantages of simple operation, the accuracy cannot satisfy the requirements of the experiment, and the sample concentration has high requirements, which are relatively difficult to attain, and the volumetric analysis is poor in sensitivity.

Comment (C) 11: The authors should state the primary goal, novelty, and contribution of this study to the field (at the end of the Introduction section).

Response (R): Thank you for your kind comments. Please see line 84-86 in page 2 for details.

The main objective of this experiment is to study the felodipine by CdTe quantum spot fluorescence spectrometry, and to provide a new idea and method for the simple and accurate determination of felodipine in tablets.

Comment (C) 12: The Method section should be revised. It was not well prepared. For example, in section 2.3.1- Selection of the felodipine dosage, it was described that “The effect of the FEL dosage on the fluorescence quenching degree was investigated.” This sentence is vague and insufficient. It is required to clarify the FEL dosage and how to investigate the “fluorescence quenching degree.” Similar to other parts of the Method section.

Response (R): Thank you for your kind comments. In order to display the content of materials and methods more clearly, we have modified this part. Please see line 88-127 in page 2-3 for details.

  1. Materials and Methods

2.1. Synthesis of CdTe quantum dots

First, 0.120 g of CdCl2·2.5 H2O and 72.7 μL of thioglycolic acid (200 mL of high-purity water) were added to a round-bottom flask, the pH value was adjusted to approximately 11.0 with 1.00 mol/L NaOH solution, and the solution was passed through high-purity nitrogen for 30 min. Then, 210 μL of the newly prepared sodium telluride solution was added under nitrogen protection. The color of the solution immediately turned yellow, and the CdTe quantum dot precursor was obtained. The original solution was transferred to a hydrothermal synthesis reactor lined with Teflon and heated in an oven at 140 °C for different times to obtain transparent CdTe quantum dot solutions of different colors [21-22].

2.2. Influence of synthesis conditions on CdTe quantum dots

Different concentrations of CdTe QD solutions (5×10-5 mol/L, 1.0×10-4 mol/L, 1.5×10-4 mol/L, 2.0×10-4 mol/L, 2.5×10-4 mol/L, 3.0×10-4 mol/L, 3.5×10-4 mol/L) were screened to explore the fluorescence intensity of the system (F2500, Hitachi Manufacturing Co., LTD.).

We selected the same concentration, dosages of 0.2 mL, 0.4 mL, 0.6 mL, 0.8 mL, 1.0 mL, 1.2 mL, and 1.4 mL of CdTe QDs, which influenced the fluorescence intensity of the system, and the best dosage of CdTe QDs was determined. When heated at 100 °C, 120 °C, 140 °C, and 160 °C for the same times, the fluorescence intensity of CdTe QDs was determined from the fluorescence emission spectra. The best heating time was determined from the fluorescence emission spectra of CdTe quantum dots at different reaction times (30 min, 60 min, 90 min, 120 min). Control feeding ratio N(Cd2+): N (Te2-): N (TGA)= 2:1:4, heating temperature 140℃, heating time 60 min, the CdTe quantum dots were analyzed by the effect of pH values of reaction precursor solution 8, 9, 10, 11, 12 on the fluorescence properties.

2.3. Structural characterization of CdTe quantum dots

The UV–visible spectrum (UV-2550) was obtained for CdTe QDs heated at 140℃ for 60 min. The particulate morphology of CdTe QDs was observed on TEM (HITACHI-800). CdTe QDs were characterized by Fourier transform infrared spectroscopy (BRUKER VERTEX70). CdTe QDs were characterized by X-ray diffraction (BRUKER D8 ADVANCE).

2.4. Interaction of CdTe quantum dots with felodipine

The fluorescence spectrum of the system with the addition of felodipine was analyzed by a fluorescence spectrophotometer. The effect of the FEL dosage on the fluorescence quenching degree was investigated. Felodipine cannot stably exist in strong acid and alkali environments, and it is easier to decompose in strong acid environments. Therefore, this experiment used weak alkaline buffer Tris-HCl to investigate the quenching degree of the fluorescence intensity of the system when the pH value of the buffer was 7.3, 7.6, 7.9, 8.2, 8.5, 8.8 and 9.1. The effect of the amount of buffer solution on the fluorescence quenching degree of the system was investigated. The quenching of the fluorescence intensity of the system for different reaction time lengths was detected.

Comment (C) 13: The authors should clarify the number of replications per experiment or measurement and how presented data are presented. Is there any statistical test?

Response (R): Thank you for your kind comments. Thank you for your kind comments. According to the experimental data, we have modified the picture. The details are as follows.

Line 66-67 in page 4: Figure 1. Influence of (A) the quantum dot concentration; (B) the amount of CdTe QDs;

Line 236-242 in page 7: Figure 4. Effect of (A) the felodipine dosage; (B) the pH of the buffer solution; (C) the amount of buffer solution; (D) the reaction time.

Line 245-246 in page 8: Figure 6. Standard working curve

Comment (C) 14: Sections 3 and 4 should be combined. The authors should rearrange the data and discussion.

Response (R): Thank you for your kind comments. Sections 3 and 4 have been combined.  Please see line 125-285 in page 3-10 for details.

  1. Results and discussion

3.1. Influence of synthesis conditions on CdTe quantum dots

3.2. Structural characterization of CdTe quantum dots

3.3. Interaction of CdTe quantum dots with felodipine

3.4. Methodological investigation

3.4.1.Precision

3.4.2. Accuracy

3.4.3. Influence of the coexisting substances

3.4.4. Sample content determination

Comment (C) 15: Both FEL and felodipine were used. The authors should keep using the abbreviation consistently.

Response (R): Thank you for your kind comments. FEL refers to felodipine. In order not to affect the expression of the content of the article, we have made marks in the introduction. Please see line 21 in page 1 for details. the felodipine (FEL) dosage was 1.0 mL,

Reviewer 2 Report

Lv et al. proposed a fluorescent probe for the detection of felodipine by CeTe quantum dots. Compared with the first version, the manuscript is improved greatly. I think it can be further improved based on the following comments.

  • Title: Title is still confusing. “Felodipine determination by a CdTe quantum dots-based fluorescent probe” for reference.
  • Abstract: Abstract is confusing. It is not suitable to use “in this review” to describe a research article. It would be “in this work”. Moreover, what is “ideal examples”? What is “handle gauge felodipine”? I suggested to rewrite the abstract by referring to the conclusion section.
  • Materials and methods: No information about the detection process.
  • Figures 1,2,6,9,10,11 didn’t show error bars.
  • I cannot understand why the authors need to investigate the dosage of felodipine? What is the difference between figure 6 and figure 11?
  • Apart from the linear range, the limit of detection should be investigated and given in this manuscript, please referring to Biosensors, 2021, 11, 6; Journal of Hazardous Materials, 2017, 321, 417-423
  • To avoid confusion, I suggested to separate the preparation of fluorescent probe and the detection of felodipine into two parts in the Materials and Methods and Results section.

Author Response

 April 28, 2022

Dear Editor,

We would like to submit the revised manuscript (micromachines-1710222) entitled “Felodipine determination by a CdTe quantum dots-based fluorescent probe” for a further review and publication in the journal Micromachines. Based on the reviewers’ comments, we have made significant revisions in a point-to-point manner (see below for details). We hope you will find that the revised manuscript now meets the standards of the journal.

Thank you very much for your consideration. We look forward to your favorable decision.

Sincerely,

Jiang Wu, Dr.

School of Stomatology, Jiamusi University, Heilongjiang 154002, China.

Email: wujiangwj0126@163.com.

Tel.: +86–454–8625581. 

Response to the comments from reviewer:

First, we would like to thank all reviewers for their critical examination of our manuscript. Their comments are very helpful for revising the manuscript. We have made significant changes to address their concerns. The issues they raised and our responses are outlined below.

Referee #2

Comment (C) 1: Title: Title is still confusing. “Felodipine determination by a CdTe quantum dots-based fluorescent probe” for reference.

Response (R): Thank you for your kind comments. We've changed the title to “Felodipine determination by a CdTe quantum dots-based fluorescent probe”. Please see line 2-3 in page 1 for details.

Comment (C) 2: Abstract: Abstract is confusing. It is not suitable to use “in this review” to describe a research article. It would be “in this work”. Moreover, what is “ideal examples”? What is “handle gauge felodipine”? I suggested to rewrite the abstract by referring to the conclusion section.

Response (R): Thank you for your kind comments. We have revised the content of the abstract part of the article. Please see line 11-27 in page 1 for details.

Abstract: In this work, a CdTe quantum dots-based fluorescent probe was synthesized to determine the felodipine. The synthesis conditions, the structural characterization, and the interaction conditions with felodipine of CdTe quantum dots were carried out by the fluorescence spectrophotometer, the fourier transform infrared spectroscopy (FT-IR), the X-ray diffraction (XRD), the UV–visible spectrum, and TEM. CdTe QD concentration was 2.0 × 10-4 mol/L. The amount of quantum dots controlled in the experiment was 0.8 mL. Control feeding ratio N(Cd2+): N (Te2-): N (TGA)= 2:1:4, heating temperature 140 ℃, heating time 60 min, the pH of the QD precursor was adjusted to 11 for subsequent experiments. The UV–visible spectrum showed that the emission wavelength of CdTe quantum dots at 545 nm was the strongest and symmetric. The particle size of the synthesized quantum dots was approximately 5 nm. In the interaction of CdTe quantum dots with felodipine, the felodipine (FEL) dosage was 1.0 mL, the optimal pH value of tris-HCl buffer was 8.2, the amount of buffer was 1.5 mL, and the reaction time was 20 min. The standard curve of felodipine was determined under the optimal conditions of synthesis conditions on CdTe quantum dots and reaction of CdTe quantum dots with Felodipine. The linear equation is Y =3.9448x+50.068, the correlation coefficient R2=0.9986, and the linear range is 5× 10-6-1.1 ×10-4 mol/L. A CdTe quantum dots-based fluorescent probe was successfully constructed, which could be used to determine the felodipine tablets content.

Comment (C) 3: Materials and methods: No information about the detection process.

Response (R): Thank you for your kind comments. We have modified this part. Please see line 88-127 in page 2-3 for details.

  1. Materials and Methods

2.1. Synthesis of CdTe quantum dots

First, 0.120 g of CdCl2·2.5 H2O and 72.7 μL of thioglycolic acid (200 mL of high-purity water) were added to a round-bottom flask, the pH value was adjusted to approximately 11.0 with 1.00 mol/L NaOH solution, and the solution was passed through high-purity nitrogen for 30 min. Then, 210 μL of the newly prepared sodium telluride solution was added under nitrogen protection. The color of the solution immediately turned yellow, and the CdTe quantum dot precursor was obtained. The original solution was transferred to a hydrothermal synthesis reactor lined with Teflon and heated in an oven at 140 °C for different times to obtain transparent CdTe quantum dot solutions of different colors [21-22].

2.2. Influence of synthesis conditions on CdTe quantum dots

Different concentrations of CdTe QD solutions (5×10-5 mol/L, 1.0×10-4 mol/L, 1.5×10-4 mol/L, 2.0×10-4 mol/L, 2.5×10-4 mol/L, 3.0×10-4 mol/L, 3.5×10-4 mol/L) were screened to explore the fluorescence intensity of the system (F2500, Hitachi Manufacturing Co., LTD.).

We selected the same concentration, dosages of 0.2 mL, 0.4 mL, 0.6 mL, 0.8 mL, 1.0 mL, 1.2 mL, and 1.4 mL of CdTe QDs, which influenced the fluorescence intensity of the system, and the best dosage of CdTe QDs was determined. When heated at 100 °C, 120 °C, 140 °C, and 160 °C for the same times, the fluorescence intensity of CdTe QDs was determined from the fluorescence emission spectra. The best heating time was determined from the fluorescence emission spectra of CdTe quantum dots at different reaction times (30 min, 60 min, 90 min, 120 min). Control feeding ratio N(Cd2+): N (Te2-): N (TGA)= 2:1:4, heating temperature 140℃, heating time 60 min, the CdTe quantum dots were analyzed by the effect of pH values of reaction precursor solution 8, 9, 10, 11, 12 on the fluorescence properties.

2.3. Structural characterization of CdTe quantum dots

The UV–visible spectrum (UV-2550) was obtained for CdTe QDs heated at 140℃ for 60 min. The particulate morphology of CdTe QDs was observed on TEM (HITACHI-800). CdTe QDs were characterized by Fourier transform infrared spectroscopy (BRUKER VERTEX70). CdTe QDs were characterized by X-ray diffraction (BRUKER D8 ADVANCE).

2.4. Interaction of CdTe quantum dots with felodipine

The fluorescence spectrum of the system with the addition of felodipine was analyzed by a fluorescence spectrophotometer. The effect of the FEL dosage on the fluorescence quenching degree was investigated. Felodipine cannot stably exist in strong acid and alkali environments, and it is easier to decompose in strong acid environments. Therefore, this experiment used weak alkaline buffer Tris-HCl to investigate the quenching degree of the fluorescence intensity of the system when the pH value of the buffer was 7.3, 7.6, 7.9, 8.2, 8.5, 8.8 and 9.1. The effect of the amount of buffer solution on the fluorescence quenching degree of the system was investigated. The quenching of the fluorescence intensity of the system for different reaction time lengths was detected.

Comment (C) 4: Figures 1,2,6,9,10,11 didn’t show error bars.

Response (R): Thank you for your kind comments. According to the experimental data, we have modified the picture. The details are as follows.

Line 66-67 in page 4: Figure 1. Influence of (A) the quantum dot concentration; (B) the amount of CdTe QDs;

Line 236-242 in page 7: Figure 4. Effect of (A) the felodipine dosage; (B) the pH of the buffer solution; (C) the amount of buffer solution; (D) the reaction time.

Line 245-246 in page 8: Figure 6. Standard working curve

Comment (C)5: I cannot understand why the authors need to investigate the dosage of felodipine? What is the difference between figure 6 and figure 11?

Response (R): Thank you for your kind comments. In the interaction of CdTe quantum dots with felodipine, the fluorescence quenching degree of the system reached the maximum when the dosage was 1.0 mL and tended to weaken when the dosage exceeded 1.0 mL (Fig. 4A). Therefore, the FEL dosage was controlled to be 1.0 mL in the experiment. And figure 6 showed that the standard curve of felodipine was determined under the optimal conditions of synthesis conditions on CdTe quantum dots and reaction of CdTe quantum dots with Felodipine. The linear equation is Y =3.9448x+50.068, the correlation coefficient R2=0.9986, and the linear range is 5× 10-6-1.1 ×10-4 mol/L.

Comment (C) 6: Apart from the linear range, the limit of detection should be investigated and given in this manuscript, please referring to Biosensors, 2021, 11, 6; Journal of Hazardous Materials, 2017, 321, 417-423

Response (R): Thank you for your kind comments. We have modified the structure, picture format and content of the paper according to the above two papers, and quoted the above literatures. Please see line 209 in page 6 for details. [23-24]

  1. Xiaohong Zhou, Abdul Ghaffar Memon , Weiming Sun, et al. Fluorescent Probe for Ag+ Detection Using SYBR GREEN I and C-C Mismatch. Biosensors, 2020; Volume 11, pp. 6. doi: 10.3390/bios11010006
  2. Abdul Ghaffar Memon, Xiaohong Zhou a, Jinchuan Liu, et al. Utilization of unmodified gold nanoparticles for label-free detection of mercury (II): Insight into rational design of mercury-specific oligonucleotides. Journal of Hazardous Materials, 2017; Volume 321, pp. 417–423. doi: 10.1016/j.jhazmat.2016.09.025

Comment (C) 6: To avoid confusion, I suggested to separate the preparation of fluorescent probe and the detection of felodipine into two parts in the Materials and Methods and Results section.

Response (R): Thank you for your kind comments. We have separated the preparation of fluorescent probe and the detection of felodipine into three parts in the Materials and Methods and Results section. There are influence of synthesis conditions on CdTe quantum dots, structural characterization of CdTe quantum dots, and interaction of CdTe quantum dots with felodipine.

Please see line 88-127 in page 2-3 for details.

  1. Materials and Methods

2.1. Synthesis of CdTe quantum dots

2.2. Influence of synthesis conditions on CdTe quantum dots

2.3. Structural characterization of CdTe quantum dots

2.4. Interaction of CdTe quantum dots with felodipine

Please see line 125-246 in page 3-8 for details.

  1. Results

3.1. Influence of synthesis conditions on CdTe quantum dots

3.2. Structural characterization of CdTe quantum dots

3.3. Interaction of CdTe quantum dots with felodipine

Round 2

Reviewer 1 Report

Review comments on micromachines-1710222v2: Felodipine determination by a CdTe quantum dots-based fluorescent probe

The authors appropriately revised the manuscript according to previous comments. The efforts are highly appreciated. Since the manuscript is in a better form now, this reviewer believe that it is possible for publication in micromachines. However, there are still several issues to consider as follows. The authors should revise the manuscript one more time and improve it for acceptance.

  1. The authors should find and include previous methods (UV-Vis and HPLC) to determine felodipine in bulk materials or any other dosage forms (e.g., tablets and capsules). In lines 67-81, there is only one previous study reporting the HPLC method (ref. #19); however, it is used for felodipine analysis in human plasma. After that, in the results and discussion, the authors should compare their devised method and the previous methods regarding linear range and LOD/ LOQ if possible.
  2. Section 2.3- Structural characterization of CdTe quantum dots: some details of the described methods should be included, such as range of UV-Vis spectrum, sample preparation for TEM, scan range, scan number, and resolution for FTIR. As QDs are in the solution state, please clarify the sample preparation for FTIR and XRD.
  3. Figures 1A, 1B, 4A, 4B, 4C, 4D, 5, and 6 should be improved because their resolution and quality are low. Where relevant, please mention the number of replicates and data presentation (means ± SDs).
  4. Figure 2C should be replaced by the original photos with better quality.
  5. Figure 2D: the X-axis of wavelength (nm) is incorrect. Please revise it.
  6. FEL and felodipine: according to the journal guidelines, after abbreviating felodipine by FEL, the authors should use FEL consistently and should not use felodipine.
  7. Does TGA mean thioglycolic acid? If so, please clarify in line 87.

Author Response

                                                    May 6, 2022

Dear Editor,

We would like to submit the revised manuscript (micromachines-1710222) entitled “Felodipine determination by a CdTe quantum dot-based fluorescent probe” for a further review and publication in the journal Micromachines. Based on the reviewers’ comments, we have made significant revisions in a point-to-point manner (see below for details). We hope you will find that the revised manuscript now meets the standards of the journal.

Thank you very much for your consideration. We look forward to your favorable decision.

Sincerely,

Jiang Wu, Dr.

School of Stomatology, Jiamusi University, Heilongjiang 154002, China.

Email: wujiangwj0126@163.com.

Tel.: +86–454–8625581.

Response to the comments from reviewers:

First, we would like to thank all reviewers for their critical examination of our manuscript. Their comments are very helpful for revising the manuscript. We have made significant changes to address their concerns. The issues they raised and our responses are outlined below.

Referee #1

Comment (C) 1: The authors should find and include previous methods (UV-Vis and HPLC) to determine felodipine in bulk materials or any other dosage forms (e.g., tablets and capsules). In lines 67-81, there is only one previous study reporting the HPLC method (ref. #19); however, it is used for felodipine analysis in human plasma. After that, in the results and discussion, the authors should compare their devised method and the previous methods regarding linear range and LOD/ LOQ if possible.

Response (R): Thank you for your kind comments.

Please see line 67-74 in page 2 for details (marked in red).

At present, few studies have been reported on the use of RP-HPLC [18], UV–VIS [19], and HPTLC [20] for the determination of FEL in bulk drugs or pharmaceutical preparations. Although UV–visible spectrophotometry has the advantages of simple operation, the accuracy cannot satisfy the requirements of experiments, and the sample concentration has high requirements, which are relatively difficult to attain, and the volumetric analysis is poor in sensitivity. HPLC has high stability and specificity, low detection and quantification limits, and is suitable for the quantitative determination of trace lodipine and its related substances [21-22].

Please see line 257-263 in page 7 for details (marked in red).

In the literature, the micellar liquid chromatography-fluorescence detection method for the determination of FEL in tablets exhibited a linear range of 0.05-15 ug/mL, with a lower LOD of 0.011 ug/ mL and a lower LOQ of 0.032 ug/mL [29]. FEL can interact with quantum dots, which can quench the fluorescence intensity of the system. The results show a good linear relationship between the decrease in fluorescence intensity and the concentration of FEL in a specific range, which formed the basis of fluorimetry to determine the FEL content.

Comment (C) 2: Section 2.3- Structural characterization of CdTe quantum dots: some details of the described methods should be included, such as range of UV-Vis spectrum, sample preparation for TEM, scan range, scan number, and resolution for FTIR. As QDs are in the solution state, please clarify the sample preparation for FTIR and XRD.

Response (R): Thank you for your kind comments.

Please see line113-131 in page 3 for details (marked in red).

The UV–visible spectrum (UV/Vis-265) was obtained for CdTe QDs by regulating the concentration of quantum dots at 1.0×10-4 mol/L, in the wavelength range of 800-200 nm, and the absorption spectrum was determined by quartz cuvette absorbance.

The concentration of the quantum dot solution was adjusted to 1.0×10-4 mol/L, and the fluorescence spectrum was detected by a fluorescence spectrophotometer (970 CRT) .

The appropriate amount of methanol was added to the CdTe solution, and fully mixed by ultrasound. The solution was centrifuged with a high-speed centrifuge and the bottom precipitate was taken, washed with methanol and centrifuged three times. The precipitate was dried in a vacuum drying oven. The prepared quantum dot solution was evaporated to 1/5 of its original volume by a rotary evaporator. After centrifugation and washing, the solution was vacuum dried at 30°C for 6 h. The CdTe QDs were characterized by Fourier transform infrared spectroscopy (Nexus-470), and the resolution was better than 0.4 cm-1. The CdTe QD powder obtained after centrifugal drying was characterized by X-ray diffraction (BRUKER D8 ADVANCE) with a Cu target Kα-ray tube (λ=0.1541 nm) at 40 kV and a scanning range of 10°–80°. The QDs obtained after centrifugal drying were dispersed into anhydrous ethanol solution (0.001-0.01 mg/mL) by ultrasound or a cell crusher until the particles were invisible to the naked eye. A pipette gun was used to drop 40 µL of the dispersion onto the front of a TEM copper mesh in two batches, which was dried at room temperature for TEM observation (HITACHI-800).

Comment (C) 3: Figures 1A, 1B, 4A, 4B, 4C, 4D, 5, and 6 should be improved because their resolution and quality are low. Where relevant, please mention the number of replicates and data presentation (means ± SDs).

Response (R): Thank you for your kind comments. Figures 1A, 1B, 4A, 4B, 4C, 4D, 5, and 6 have been improved.

Please see line181-182 in page 5 for details (marked in red). Figure 1. Influence of (A) the quantum dot concentration; (B) the amount of CdTe QDs

Please see line 266-274 in page 8-9 for details (marked in red).

Figure 4. Effects of (A) FEL dosage; (B) pH of the buffer solution; (C) amount of buffer solution; and (D) reaction time.

Figure 5. The structure of the complex formed by thioglycolic acid and Cd2+.

Figure 6. Standard working curve.

The number of replicates and data presentation (means ± SDs) have been supplemented.

Please see line158 in page 4 for details (marked in red) (225.338±0.15326, n=3)

Please see line165 in page 4 for details (marked in red)(241.406±0.2267, n=3)

Please see line229 in page 7 for details (marked in red) (245.164±0.0375, n=3)

Please see line236 in page 7 for details (marked in red) (262.335±0.0375, n=3)

Please see line244-246 in page 7 for details (marked in red) When 1.5 mL of buffer solution was controlled in the experiment, the fluorescence intensity was 258.708±0.02003 (n=3).

Please see line250 in page 7 for details (marked in red) (247.768±0.01682, n=3)

Comment (C) 4: Figure 2C should be replaced by the original photos with better quality.

Response (R): Thank you for your kind comments. Please see line206-207 in page 6 for details (marked in red). (C) Transmission electron microscopy of CdTe quantum dots;

Comment (C) 5: Figure 2D: the X-axis of wavelength (nm) is incorrect. Please revise it.

Response (R): Thank you for your kind comments. According to the experimental data, we have modified the picture. The details are as follows. Please see line206-207 in page 6 for details (marked in red). (D) Infrared spectra of CdTe quantum dots;

Comment (C) 6: FEL and felodipine: according to the journal guidelines, after abbreviating felodipine by FEL, the authors should use FEL consistently and should not use felodipine.

Response (R): Thank you for your kind comments. We have abbreviated felodipine by FEL.

The modified FEL has been marked in red in the paper.

Comment (C) 7: Does TGA mean thioglycolic acid? If so, please clarify in line 87.

Response (R): Thank you for your kind comments. We have labeled the TGA on line 87. Please see line 87 in page 2 for details (marked in red).

First, 0.120 g of CdCl2·2.5 H2O and 72.7 μL of thioglycolic acid (TGA) (in 200 mL of

Reviewer 2 Report

I suggested the authors to correct their English expression by a native speaker or an English-polishing company.

Author Response

                                                    May 6, 2022

Dear Editor,

We would like to submit the revised manuscript (micromachines-1710222) entitled “Felodipine determination by a CdTe quantum dot-based fluorescent probe” for a further review and publication in the journal Micromachines. Based on the reviewers’ comments, we have made significant revisions in a point-to-point manner (see below for details). We hope you will find that the revised manuscript now meets the standards of the journal.

Thank you very much for your consideration. We look forward to your favorable decision.

Sincerely,

Jiang Wu, Dr.

School of Stomatology, Jiamusi University, Heilongjiang 154002, China.

Email: wujiangwj0126@163.com.

Tel.: +86–454–8625581.

Response to the comments from reviewers:

First, we would like to thank all reviewers for their critical examination of our manuscript. Their comments are very helpful for revising the manuscript. We have made significant changes to address their concerns. The issues they raised and our responses are outlined below.

Referee #2

Comment (C) 1: I suggested the authors to correct their English expression by a native speaker or an English-polishing company.

Response (R): Thank you for your kind comments. We have revised the English expression of the paper by a professional English editing company. The modified details has been marked in red in the paper.

Round 3

Reviewer 1 Report

The manuscript was appropriately revised and can be accepted as is.

This manuscript is a resubmission of an earlier submission. The following is a list of the peer review reports and author responses from that submission.

Round 1

Reviewer 1 Report

Authors report the synthesis of cadmium telluride quantum dots and their application to the quantification of felonious based in the quenching of the fluorescence. The article presents several important deficiencies  and I do not recommend its publication. I encourage authors to complete address the comment below and improve their work

Title

- Title does not represent the content of the article since the mechanism is not demonstrated and no experimental data is presented to support the mechanism of interaction between CdTe quantum dots and felodipine

Abstract

  • Abstract should be re-elaborated to highlight the main findings and the relevance of the work. English must be improved, in particular in the abstract.

    Introduction

  • Please, precise the term Double hydrogen pyridine. Do you mean 1,4-dihydropyridine? Double hydrogen pyridine is not found in Scifinder and in Google only 4 entries: 2 in Chinese (one unaccessible), 1 from a Chinese company offering amlodipine tablets and 1 paper (doi: 10.1016/j.theriogenology.2017.07.012) where the term in mentioned only once. None of them gives a definition.

  • Please, reference the analytical methods for the determination of felodipine and report the main limitations to contextualize the importance of the paper

Instrument and Materials

  • Please, move the second paragraph describing the synthesis of the nanoparticles to correct location (i.e. methods) and include a title (for example: “Synthesis of CdTe quantum dots”). Additionally, report the concentration or number of moles of the reactant, in particular of the newly prepared sodium telluride solution, as well as the time at 140 C.

Experimental methods

  • The label of section 3 should not be “Experimental methods” but “Results”

  • Particles are not characterized. Please, include size, Z-potential, XPS, quantum yield, uv-vis and fluorescence spectra). The analysis of the uv-vis and fluorescence spectra is important to understand the quenching (please, check DOI 10.1007/s10953-009-9397-0)

  • Authors report a linear relationship between fluorescence and concentration of felodipine but they do not include any data to support it (section 3.1). Please, include a plot fluorescence vs concentration of felodipine showing the linear relationship and the R2 value, as in figure 9. Authors should analyzed their data after reading the article DOI: 10.1007/s11095-015-1725-z.

  • Authors study the effect of the concentration of CdTe Qdots on the fluorescence and observe quenching. However, at high concentration the fluorescence raises. Please, comment. On the other hand the concentration is presented as mol/L, and the calculations are “based on Cd2+”. Please, comment. How do you determine the molecular weight of you nanoparticles? Please, recall that the mass will be dependent on the size and although the size distribution (that you do not report) may be narrow, a population of particles with different sizes (i.e. molecular mass) will be always present. On the other hand you do not report the content of Cd2+

  • The influence of the dosage is confusing (section 3.3). What was the concentration of Qdots solution used to plot figure 3? If you plot it not as a function of volume but of concentration, you should get Figure 2. What is the rational behind this study? In oder to set up you assay you must define the concentration of QDots. Please, recall that with the information that you provide the experiment can not be reproduced: you report the volumes of an unknown solution that is added to a unknown volume of other solution/s (water, buffer?) and conclude that “when the dosage was 0.8-1 mL, the fluorescence quenching degree was strong and stable”

  • In section 3.4 the same lack of information is present. Please, report you results as concentrations instead of volumes of unknown solutions. Comment the fact that you are obtaining the same plot independently that you study concentration of QDs, volume of QDs or volume of FEL.

  • Figure 5 is not correct: the structure on the right side has two sulfurs. Please, provide details (i.e. experimental results or references) to support the proposed mechanism of the enhancement of the fluoresce.

  • Please, detail the experiment that you report in sections 3.5 and 3.6. Are you measuring the quenching as a function of pH with or without the analyte?

  • The study on the amount of buffer (section 3.6) suffer the same problems as those for section 3.3. What was the final pH? If, as expected, it was that of the buffer, what you are plotting in Fig 7 is the ionic strength of the system. Please, comment.

  • Please, comment on the reaction responsible for the lag time to achieve the maximum quenching.

  • Section 3.8: please, report the optimum experimental conditions.

  • Section 4.2. Please, describe the experiment

  • Section 4.3. Please, describe the experiment

  • Section 4.4: the hypothesis is not supported by any experimental data. At the pH of the assay, the carboxylic groups are expected to be deprotonated. Why don´t they interact with the nitrogen instead of the oxygen of the carbonyl group from the ester? Please, check the articles DOI 10.1007/s10953-009-9397-0 and DOI 10.1007/s11095-015-1725-z and comment on how your hypothesis based on non covalent interactions can explain the lag time to reach the maximum quenching.

Reviewer 2 Report

Nie et al. proposed cadmium telluride (CdTe) quantum dots as fluorescent prob for felodipine detection. This is an interesting work however with poor writing. I can not clearly catch up with the main ideas of this work. For example, the authors used “Mechanism of interaction between CdTe quantum dots and fe-2 lodipine” as the title. However, they did not provide strong evidence for the mechanism identification between of them. When I read the abstract, still a lot of confusing and repeatable expression existed. So I strongly suggested the authors ask help from native speakers to check through the whole manuscript for next submission.

Reviewer 3 Report

Dear authors,

I recommend that you conduct serious research and ask for help in writing scientific data in English

Line 12: constructed for chemist is synthesizes, the actual sample should be precisely mentioned !

Line 14: “It provides a new reliable method for the determination of felodipine. ”

Line 16-18:  ”A convenient, accurate and efficient new method was established for the determination of felodipine content. This method was used as a fluorescent probe for the determination of felodipine content, and the experimental results were satisfactory.” The same statement in two sentences!

Line 24-27 – first sentence of the introduction section is too long and grammatical correctness is critically lacking ! needs rephrase !

Line 33 – Introduction is the part where the authors are introducing the state of art of the subject and cannot conclude anything !

Line 29-35 are not supported by bibliographic references.

Line 37-40 – again a meaningless sentence in which the authors mix scientific notions about felodipine. Unfortunately, the article needs a considerable revision of the English writing.

Line 60 – too many references indicated

Line 62 – UV with uppercase

Line 67-71 – a completely grammatically incorrect sentence

Line 91-97 - another incomprehensible phrase that lacks scientific and grammatical correctness.

All graphs in Fig. 2- Fig. 8 are not supported by reproducibility experiments, the standard deviation being absent

Reviewer 4 Report

Review comments on micromachines-1584163: Mechanism of interaction between CdTe quantum dots and felodipine

The manuscript described the evaluation of CdTe quantum dots on the quantitation of felodipine. The authors provided some data to highlight and interpret the interaction between CdTe quantum dots and felodipine. However, the manuscript was not well prepared. It did not follow the journal guideline. There was no Method section. The data were insufficient to confirm the successful development of a new method to quantify felodipine. Therefore, this manuscript was not suitable for publication in the current form. The detailed comments are as follows.

  1. The abstract was vague and insufficient. The authors should provide primary points of methodology and results.
  2. The Introduction section lacked references. The authors should cite relevant references to support the statements in lines 28-36 and 61-73. On the other hand, in line 60, the authors cited 12 references at one (ref. 6-17). The authors should only cite the relevant and critical ones.
  3. The paragraph in lines 61-73 was vague and insufficient: the authors should expand this part by providing and discussing previous studies to detect felodipine.
  4. The novelty and contribution should be clearly stated in the Introduction section.
  5. There were no relevant references for the method of CdTe quantum dot preparation (lines 78-85). Importantly, please transform this part to complete sentences. They are currently incomplete sentences.
  6. Sections 3 and 4 are Results and discussion. There was no Method section. All the methodologies were not described and cited.
  7. Data in Figures 2, 3, 4, 6, 7, and 8 should be means ± SDs.
  8. The quantum dot can interact with other molecules, which interferes with the quantitation of felodipine. The specificity of this method should be evaluated.
  9. LOD and LOQ of the method should be identified and compared with the conventional concentration of felodipine in human blood.
  10. The validation of the method (specificity, accuracy, precision, etc.) should be conducted using a matrix (human or animal blood/ plasma) to show its ability to apply in clinical or experimental samples.